# EVI2B Is a Prognostic Biomarker and Is Correlated with Monocyte and Macrophage Infiltration in Osteosarcoma Based on an Integrative Analysis

**DOI:** 10.3390/biom13020327

**Published:** 2023-02-08

**Authors:** Zhenlin Dai, Zheqi Liu, Rong Yang, Wei Cao, Tong Ji

**Affiliations:** 1Department of Oral and Maxillofacial & Head and Neck Oncology, Shanghai Ninth People’s Hospital, Shanghai Jiao Tong University School of Medicine, Shanghai 200011, China; 2Shanghai Key Laboratory of Stomatology & Shanghai Research Institute of Stomatology, National Clinical Research Center of Stomatology, Shanghai 200011, China

**Keywords:** osteosarcoma, EVI2B, macrophage infiltration, gene set enrichment analysis, prognostic model

## Abstract

Osteosarcoma (OS) is the most common malignant bone tumor. However, treatment strategies have not changed over the past 30 years. The relationship between OS and the immune microenvironment may provide a basis for the establishment of novel therapeutic targets. In this study, a large-scale gene expression dataset (GSE42352) was used to identify key genes in OS. A Target-OS dataset from the Cancer Genome Atlas was used as a validation set. Ecotropic viral integration site 2B (*EVI2B*) was significantly upregulated in OS tumor samples. Differentially expressed genes (DEGs) were identified between samples with high and low *EVI2B* expression in both the test and validation cohorts. The top three functions of DEGs determined by a gene set enrichment analysis (GSEA) were chemokine signaling, cytokine–cytokine receptor interaction, and Human T-cell leukemia virus 1 infection. A prognostic prediction model including *EVI2B*, *DOCK2*, and *CD33* was constructed by a Cox regression analysis. This model indicated that *EVI2B* is an independent protective prognostic marker in OS. An analysis of immune infiltration further showed that high *EVI2B* expression levels were correlated with high levels of macrophage infiltration. Protein expression data derived from the Human Protein Atlas suggested *EVI2B* to be highly expressed in monocytes. Finally, we validated the elevated expression of *EVI2B* in OS cell lines and OS tissue samples; these results were consistent with those of the analyses of the GSE42352 and Target-OS datasets. Our integrative bioinformatics analysis and experimental results provide clear evidence for the prognostic value of *EVI2B* in OS and its close relationship with monocyte and macrophage infiltration.

## 1. Introduction

Osteosarcoma (OS) is the most common malignant bone tumor. It typically has a bimodal age distribution, with the highest incidence during adolescence, followed by a plateau, and a small peak in elderly individuals [1,2]. OS occurs disproportionately in the metaphysis of long bones, most commonly occurs in the distal femur (43%), proximal tibia (23%), or humerus (10%) [3]. The typical symptoms of OS are local swelling and pain, and the most life-threatening problems are local recurrence and distal metastasis to the lung. Current treatment strategies for OS have been in use since the mid-19th century, including intensive multidrug chemotherapy and surgical resection; these strategies have raised the 5-year survival rates to 60%, with no additional improvement in the past few decades. Patients who have metastatic or recurrent disease have a particularly low 5-year survival rate of below 30% [4,5,6,7].

OS is characterized by significant somatic copy number alterations (SCNA) and structural variation (SV), with few common features between tumors, except for mutations in the tumor suppressors RB1 and TP53 [8]. These characteristics present an extreme challenge with respect to the identification of somatic therapeutic targets for OS. Despite the genomic complexity and intratumor heterogeneity, OS is thought to be targetable and opportunities are emerging to exploit genome-level abnormalities [9,10,11]. Immunotherapy is also thought to have potential applications in OS and many immune-related therapeutic indicators, including PD-1, PD-L1, HER2, IGF2R, CTLA-4, and GD2, are highly expressed in tumor samples [12,13,14,15,16,17,18]. However, the development of new treatment strategies and biomarkers for OS has been hampered by various complications, and clinical translation is still mostly confined to pre-clinical studies [12,16,17,18,19,20]. Therefore, genome-wide analyses are needed to provide new research directions.

With the aim of exploring diagnostic and prognostic factors in OS, we used datasets from the Gene Expression Omnibus (GEO) database (GSE42352) and The Cancer Genome Atlas (TCGA, Target-OS) as test and validation sets. Based on these two datasets, we identified and further characterized EVI2B, a key gene with prognostic and diagnostic value in OS, developed a prediction model, and evaluated associations between gene expression and immune infiltration. The identification and discovery of this new prognostic marker can empower clinical decisions, and has great significance for clinical staging and predicting the outcome of clinical therapies.

## 2. Materials and Methods 

### 2.1. Data Acquisition and Differentially Expressed Gene Analysis

We found that *EVI2B* in the GSE42352 dataset was significantly differentially expressed between 15 normal samples and 103 OS samples, suggesting that this gene is a candidate biomarker for diagnosis or prognosis in OS. The samples from 103 patients with OS in GSE42352 were divided into a high-expression group (*n* = 52) and a low-expression group (*n* = 51) according to the level of *EVI2B* (setting the median value as a cut-off). Differentially expressed genes (DEGs) between the high and low *EVI2B* expression groups were identified using the limma package in R [21]. |log2(Fold Change)| > 1 and adjusted *p* < 0.05 were set as threshold values for significant DEGs. The DEG results were visualized using the pheatmap and ggplot2 packages in R. The validation cohort was derived from TCGA Target-OS (dataset https://ocg.cancer.gov/programs/target/data-matrix, accessed on 15 July 2021). Applying the same method used for the test set, 101 samples in Target-OS were divided into high expression (*n* = 51) and low expression (*n* = 50) groups according to the median *EVI2B* value.

### 2.2. Functional Analysis of DEGs

Gene Ontology (GO) terms in the three broad categories BP (biological process), MF (molecular function), and CC (cellular component) were analyzed using the enrichGO function of the clusterProfiler R package [22]. Kyoto Encyclopedia of Genes and Genomes (KEGG) analyses were performed using KOBAS 3.0 (http://kobas.cbi.pku.edu.cn accessed on 15 July 2021). Adjusted *p*-values (false-discovery rate [FDR]) of less than 0.05 were considered statistically significant. GO and KEGG results were visualized using the ggplot2 package in R. A gene set enrichment analysis (GSEA) was performed to elucidate the functional and pathway differences between the high- and low-*EVI2B* groups. The GSEA was performed using the clusterProfiler package and results were visualized using the RColorBrewer package in R.

### 2.3. Protein–Protein Interaction (PPI) Network Analysis

DEGs from GSE42352 were uploaded to STRING (https://string-db.org accessed on 15 July 2021) and a confidence value of greater than 0.4 was set for the PPI network analysis. Cytoscape (version 3.7.2) [23] was used to visualize the network. Molecular complex detection (MCODE) [24] and cytoHubba were applied to screen hub genes in the PPI network.

### 2.4. ceRNA Network

The limma package was used to obtain differentially expressed lncRNAs in the Target-OS dataset. A list of highly conserved microRNAs that were predicted targets of the lncRNAs was obtained using miRcode (http://www.mircode.org/index.php, accessed on 15 July 2021). Then, mRNA targets of miRNAs were predicted using miRDB (http://www.mirdb.org/cgi-bin/search.cgi, accessed on 15 July 2021), miRTarBase (http://mirtarbase.cuhk.edu.cn, accessed on 15 July 2021) [25], and TargetScan (http://www.targetscan.org, accessed on 15 July 2021) [26]. The intersection between mRNAs obtained from the Target-OS dataset and predicted mRNAs was obtained and used for the construction of a ceRNA network.

### 2.5. Analysis of Immune Infiltration and Correlations with EVI2B Expression

The CIBERSORT package [27] was used to compare the abundance of 22 kinds of immune cells between the high and low *EVI2B* expression groups in GSE42352, including different T cells, B cells, plasma cells, natural killer cells, and various myeloid subsets. Correlations between the expression of different immune cells and *EVI2B* in the two groups were also analyzed. The Wilcoxon rank-sum test and Spearman correlation coefficients were used to evaluate associations between *EVI2B* expression and immune cells.

### 2.6. Expression of EVI2B in Different Tissues and Immune Cells

Human Protein Atlas (HPA) (https://www.proteinatlas.org, accessed on 15 July 2021) was used to analyze the protein and mRNA expression levels of EVI2B in different tissues. Three datasets including HPA scaled dataset, Monaco scaled dataset, and Schmiedel dataset in HPA were used to analyze the expression of EVI2B in various immune cells.

### 2.7. Tissue Samples and Cell Culture

Paired fresh OS tissues and their corresponding adjacent normal tissues were collected from patients at the Ninth People’s Hospital, Shanghai Jiao Tong University School of Medicine. All patients were well informed about the study and provided written informed consent, and the process was approved by the Ethics Committee of the Ninth People’s Hospital, Shanghai Jiao Tong University School of Medicine. The OS cell lines 143B U-2 OS, Saos-2, HOS, MNNG/HOS, SJSA-1, and MG-63 were obtained from the American Type Culture Collection (Manassas, VA, USA). Fibroblast cells were cultured from adjacent normal tissues of patients with OS. All cells were cultured in 90% DMEM (Gibco, Grand Island, NY, USA) supplemented with antibiotics (1 × penicillin/streptomycin 100 U/mL, Gibco) and 10% heat-inactivated fetal bovine serum (FBS; Gibco). The cells were incubated at 37 °C in a humidified and 5% CO_2_ incubator.

### 2.8. PCR Analysis

Total RNA was extracted from cells by TRIzol reagent (Invitrogen, Thermo Scientific, Shanghai, China), and RNA was reverse-transcribed into cDNA using the Reverse Transcription Kit (QIAGEN, Valencia, CA, USA). qPCR analyses were performed with SYBR-Green (Takara, Kusatsu, Japan), and the gene expression levels were normalized against the level of *GAPDH*. The primers for *EVI2B* were as follows: forward, GTGGACACCTGAACAATACAT; reverse, TTTGTGGGGTTTGTTTGTTAC. The primers for *GAPDH* were as follows: forward, ATGACATCAAGAAGGTGGTGAAGCAGG; reverse, GCGTCAAAGGTGGAGGAGTGGGT.

### 2.9. Immunohistochemical Staining

Paraffin-embedded tissues were used for immunohistochemical staining for EVIB2 detection. The slides were dried at 60 °C, dewaxed with methanol, and rehydrated with alcohol. Then, the slides were immersed in 3% hydrogen peroxide and labeled with antibodies overnight. The anti-EVI2B (1:100; NBP1-85342) antibody was purchased from Novus (St. Charles, MO, USA).

### 2.10. Statistical Analysis

Statistical analyses were performed using R (4.0.2). *EVI2B* expression was compared between normal and tumor tissues by Wilcoxon rank-sum tests. An ROC curve was generated to evaluate the performance of *EVI2B* as a diagnostic marker using the pROC package in GSE42352. Univariate and multivariate Cox regression analyses were performed to identify risk biomarkers for OS using the Target datasets. A risk score for each patient was calculated as the sum of the scores for each parameter, obtained by multiplying the value of each biomarker and its coefficient. According to the median risk scores, patients were divided into low- and high-risk groups. Survival analyses, including associations between overall survival and disease-free survival and EVI2B and biomarkers, were performed using the survival package. All statistical analyses were two-sided, and *p*-values of <0.05 were considered significant.

## 3. Results

### 3.1. Identification of Differentially Expressed Genes

The GSE42352 dataset, including 15 normal tissues and 103 OS tissues, was chosen as the test set. *EVI2B* was significantly upregulated in the OS samples compared with normal tissues (Figure 1A). An ROC analysis revealed that *EVI2B* can be used as a marker to distinguish tumors from non-tumors in OS (Figure 1B, AUC = 0.860). Then, samples from 103 patients with OS in GSE42352 were divided into high-expression (*n* = 50) and low-expression groups (*n* = 51) according to the median *EVI2B* expression level. In total, 151 DEGs were identified in the comparison between groups (as summarized in a volcano plot and heatmap in Figure 1C,E). In the Target-OS dataset derived from TCGA, chosen as the validation set, 543 DEGs were obtained (Figure 1D,F).

### 3.2. Functional Cluster Analysis of DEGs in GSE42352

The functions of DEGs in patients with OS were predicted by GO and KEGG enrichment analyses. The top enriched GO terms in the BP, MF, and CC groups were immune response (GO:0006955), peptide antigen binding (GO:0042605), and extracellular exosome (GO:0070062), respectively (Appendix A). A KEGG pathway enrichment analysis using the KOBAS database indicated the DEGs were mainly involved in metabolic pathways, immune-related pathways, such as Th1 and Th2 cell differentiation, Th17 cell differentiation, and cancer-related pathways, such as autophagy, PI3K−Akt, MAPK, and Toll-like receptor signaling pathways (Appendix A). GSEA results are summarized in Figure 2A. The top three pathways were chemokine signaling, cytokine–cytokine receptor interaction, and Human T-cell leukemia virus 1 infection, which are related to the immune response (Figure 2B).

### 3.3. Construction of a ceRNA Network

In the Target dataset, 26 differentially expressed lncRNAs were found (Appendix A) and were visualized by a heatmap (Figure 2C). Additionally, 22 miRNAs predicted to target the lncRNAs were identified using miRcode. miRDB, miRTarBase, and TargetScan were used to predict mRNAs able to interact with the miRNAs. Finally, the intersection of the predicted mRNAs and DEGs in the Targets dataset was obtained, yielding 44 genes. Using these loci, a ceRNA network was constructed (Figure 2D).

### 3.4. PPI Network Analysis and Identification of Hub Genes

To identify hub genes among the DEGs identified in the GSE42352 dataset, a PPI network was constructed (Figure 3A). Five groups of hub genes were selected using the MCODE function in Cytoscape and the group with the highest value is shown in Figure 3B. We also used the cytoHub function to filter the key genes in the PPI network. The top 20 genes are shown in Figure 3C; these genes were similar to the hub genes obtained using MCODE.

### 3.5. Prognostic Analysis and Construction of a Prediction Model for Osteosarcoma

Kaplan–Meier curves for overall survival and disease-free survival were plotted to analyze prognosis for patients with different *EVI2B* expression levels (Figure 4A,B). Log-rank tests revealed that high levels of *EVI2B* are significantly associated with a better prognosis. We further aimed to develop a gene-signature-based risk model. First, we obtained overlapping DEGs between the GSE42352 and Target datasets, for a total of 64 genes (Figure 4C). A univariate Cox regression analysis was performed using the Target dataset to identify the top ten survival-related genes. Next, multivariate Cox regression analysis was performed to further identify genes with strongest correlations (Table 1). *EVIB2*, *DOCK2*, and *CD33* were finally selected as prognostic biomarkers. Risk scores were calculated (Figure 4D). As determined by Kaplan–Meier curves for overall survival (Figure 4E), a higher risk score was associated with a poorer prognosis in the Target dataset. In an ROC analysis of the risk model, the AUC value was 0.725, indicating that it had good accuracy for the prediction of prognosis in OS (Figure 4F).

### 3.6. Correlation between EVI2B and Infiltrating Immune Cells

The relative abundances of 22 kinds of immune cells were calculated using CIBERSORT. The levels of several immune cells differed significantly between the groups with high and low *EVI2B* expression (Figure 5A,B). In these immune cells, M2 macrophages were significantly upregulated in the low-*EVI2B*-expression group in both datasets. The correlations among immune cell types were visualized in a heatmap (Figure 5C,D). M2 macrophages were negatively correlated with activated dendritic cells and naïve CD4+ T cells (*R* < −0.5).

Since *EVI2B* is associated with a low risk for OS and may play a key role in the immune microenvironment in this cancer, we further investigated the relationship between immune cells and prognosis in 47 samples with low *EVIB2* expression. M0 macrophages, activated myeloid dendritic cells, CD4+ memory activated T cells, neutrophils, monocytes, and M2 macrophages were selected based on their significant differences between groups with low and high *EVI2B* expression. Samples were divided into high and low groups according to the infiltration abundance score calculated using CIBERSORT. Although the difference was not statistically significant (owing to the small sample size), a survival difference can be observed in the plots (Appendix A). Further studies with larger sample sizes are required to validate this finding.

### 3.7. EVI2B Expression in Immune Cells

We used three datasets in HPA to analyze the expression of EVI2B in different immune cells (Figure 6A–C) and different tissues (Appendix A). EVI2B was highly expressed in monocytes, especially in neutrophils.

### 3.8. EVI2B Is Upregulated in Osteosarcoma Cell Lines and Tissues

Six cell lines and five jaw osteosarcoma (JOS) tissue samples were used to investigate the mRNA expression of *EVI2B* by qPCR. Compared with levels in fibroblast cells, *EVI2B* expression was remarkably higher in all OS cell lines (Figure 7A). In three of five JOS tissues, the expression of *EVI2B* was upregulated (Figure 7B). To further explore the protein expression of *EVI2B*, we collected 27 JOS samples. Immunohistochemical results showed that the protein levels of EVI2B in tumor tissues were higher than those in adjacent normal tissues (Figure 7C,D).

## 4. Discussion

The most recent major breakthrough in the treatment of OS was reported in the mid-19th century, when the clinical efficacy combination of doxorubicin, cisplatin, and methotrexate was established [5]. Despite tremendous and ongoing efforts, research aimed at the development of new therapeutic strategies for patients with recurrent or metastatic OS has been relatively unproductive. Moreover, biomarkers for diagnosis or stratifying patients for specific therapeutic options are lacking. For the past few years, major advances in next-generation sequencing technology have provided extensive data for research [28,29,30,31]. In view of the difficulty in obtaining and preserving fresh OS samples, high-throughput platforms and databases are particularly promising for the detection of candidate diagnostic indicators and therapeutic targets [32,33,34,35]. We evaluated the GSE42352 GEO dataset, including 103 OS tissue samples, to eliminate heterogeneity. In this dataset, DEGs in OS of were enriched in immune-related signaling pathways, including chemokine signaling, cytokine–cytokine receptor interaction, and Human T-cell leukemia virus 1 infection. *EVI2B* (Ecotropic viral integration site 2B, CD361) was significantly upregulated in OS samples. We validated the upregulation of EVI2B at both the mRNA and protein levels in cell lines and fresh tumor tissues (using fibroblasts and adjacent normal tissues for comparison), supporting the reliability of our findings. We also constructed a ceRNA network consisting of 44 genes for OS. We then obtained 20 genes from the PPI network based on DEGs expected to contribute to prognosis in OS. Among these genes, *EVIB2, DOCK2,* and *CD33* were finally selected as prognostic biomarkers.

Tumor cells can gain a growth advantage via immune escape. Immunotherapy, a new therapeutic approach, has become one of the five pillars of tumor therapy, along with surgery, cytotoxic chemotherapy, molecularly targeted therapy, and radiation therapy [36]. Preclinical and clinical studies have advanced rapidly in the past few years. However, the lack of therapeutic methods and poor treatment efficacy in patients with recurrent or metastatic OS emphasizes the need for further analyses of the effects of immunotherapy. Many immune-related therapeutic indicators, including PD-1, PD-L1, HER2, IGF2R, CTLA-4, and GD2, are highly expressed in tumor samples [12,13,14,15,16,17,18,37]. However, the efficacy of immunotherapy in patients with unselected OS is limited [38,39,40,41]. The most highly dysregulated gene in our study, *EVI2B*, encodes a type I transmembrane protein and is embedded in intron 27b of the neurofibromatosis type 1 (*NF1*) gene; it is transcribed in the opposite direction to that of the *NF1* gene [42]. During the Ninth HLDA Workshop (HLDA9), *EVI2B* was identified as one of eighteen new cell-surface molecules expressed on B cells [43]. Its function is not yet fully understood. *EVI2B* is a downstream target of CCAAT/enhancer binding protein alpha (C/EBPα), playing an important role in myeloid differentiation and the function of hematopoietic progenitors, thereby contributing to the onset of leukemia, consistent with our findings that the expression of *EVI2B* was highest in mature granulocytes [44]. Furthermore, some studies in the field of osteoarthritis have demonstrated that some bone marrow mesenchymal stem-cell-derived cells, including neutrophils (EVI2B highly expressed in OS), can affect the differentiation or function of (OBs) or osteoclasts (OCs) [45,46,47]. OCs are considered to be highly specialized macrophages and closely related to the immune microenvironment and immunotherapy. For example, it has been shown that the presence of osteoclasts can affect the cytotoxicity of NK cells and the loss of surface receptors on osteoclasts is correlated with a decreased expansion and function of NK cells [48]. Additionally, PD-1 blockade inhibits bone cancer pain by inhibiting osteoclast formation, partly [49]. In view of the findings of this study and previous research, we speculated the dysregulation of EVI2B on immune cell may alter the bone homeostasis by affecting the generation or function of OBs or OCs, thus affecting the corresponding response to therapy and prognosis. These findings may also explain the poor efficacy of dendritic cell vaccines against OS in clinical trials [40,50,51]. In addition to *EVI2B*, we also identified *DOCK2* and *CD33* as prognostic markers for OS; these genes are important for immune regulation [52,53,54]. The discovery of these biomarkers improves our understanding of the immune microenvironment in OS and provides an important basis for the development of future immunotherapy strategies.

However, there were some inevitable limitations of our study. For example, the overexpression of *EVI2B* is associated with the postoperative recurrence of colorectal cancer [55], differing from the above research findings. Additionally, hemizygosity of genes, including *EVI2B* in the deleted region around the neurofibromin locus, may contribute to a severe phenotype in patients with NF1 [56]. Although we preliminarily explored the biological process by which *EVI2B* contributes to OS by an enrichment analysis, the detailed mechanism requires further biomedical experiments. The precise role of this gene in the immune microenvironment and its impact on the response to immunotherapy should be focuses of future research.

Furthermore, considering the obvious intra-tumoral heterogeneity of OS, the lack of multi-locus sampling data within a single tumor sample in these public databases limits the value of the biomarker and prognostic prediction model. Although we confirmed the elevated mRNA expression levels of *EVI2B* in six OS cell lines compared with levels in fibroblasts by qPCR, this difference was not as clear in the five fresh OS tissues samples. Additionally, the specific genomic characteristics of OS, including SCNV and SV, challenge the utility of the newly established biomarker. Epigenetic abnormalities in tumors were not analyzed. We were also unable to conduct sample sequencing and subsequent bioinformation analyses of a larger cohort for internal validation. Further studies are therefore still warranted.

In conclusion, genomic data for OS obtained from the GEO and TCGA databases were integrated to obtain a system-level view of gene expression profiles. We found that the upregulated hub gene *EVI2B* and several key immune-related signal pathways are involved in the development of OS and built a prognostic prediction model consisting of *EVI2B*, *DOCK2,* and *CD33*. These findings provide new insights into the genomic underpinnings of the pathogenesis of OS and potential therapeutic targets for immunotherapy.

## Figures and Tables

**Figure 1 biomolecules-13-00327-f001:**
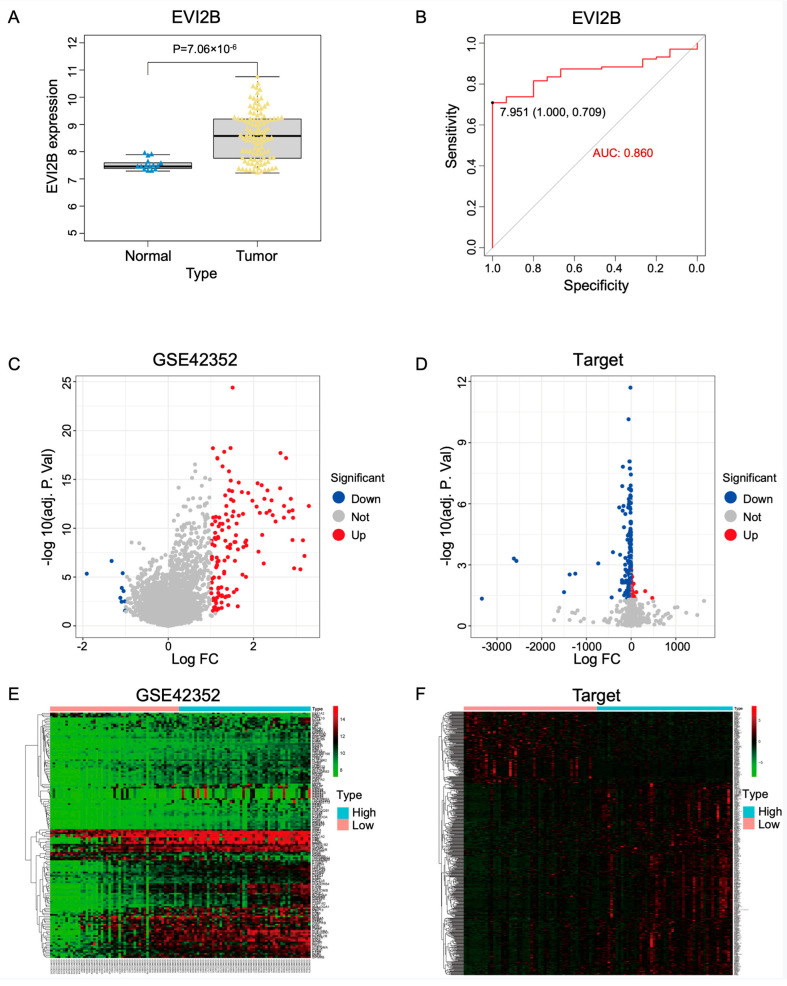
Analysis of *EVI2B* and DEGs (**A**) *EVI2B* expression was significantly lower in tumor tissues than in normal tissues. (**B**) The area under ROC curve (AUC) for *EVI2B* in the test set was used to evaluate its diagnostic performance for osteosarcoma. (**C**) The GSE42352 dataset (test set) was divided into high- and low-expression groups according to the median expression of *EVI2B*. Significant DEGs between groups are shown in a volcano plot. (**D**) The Target−OS dataset (validation set) was divided into high- and low-expression groups according to the median expression of *EVI2B*. Significant DEGs between groups are shown in a volcano plot. (**E**) DEGs between high- and low-*EVI2B*-expression groups in the GSE42352 dataset are presented in a heat map. (**F**) DEGs between high- and low-*EVI2B*-expression groups in the Target−OS dataset are presented in a heat map.

**Figure 2 biomolecules-13-00327-f002:**
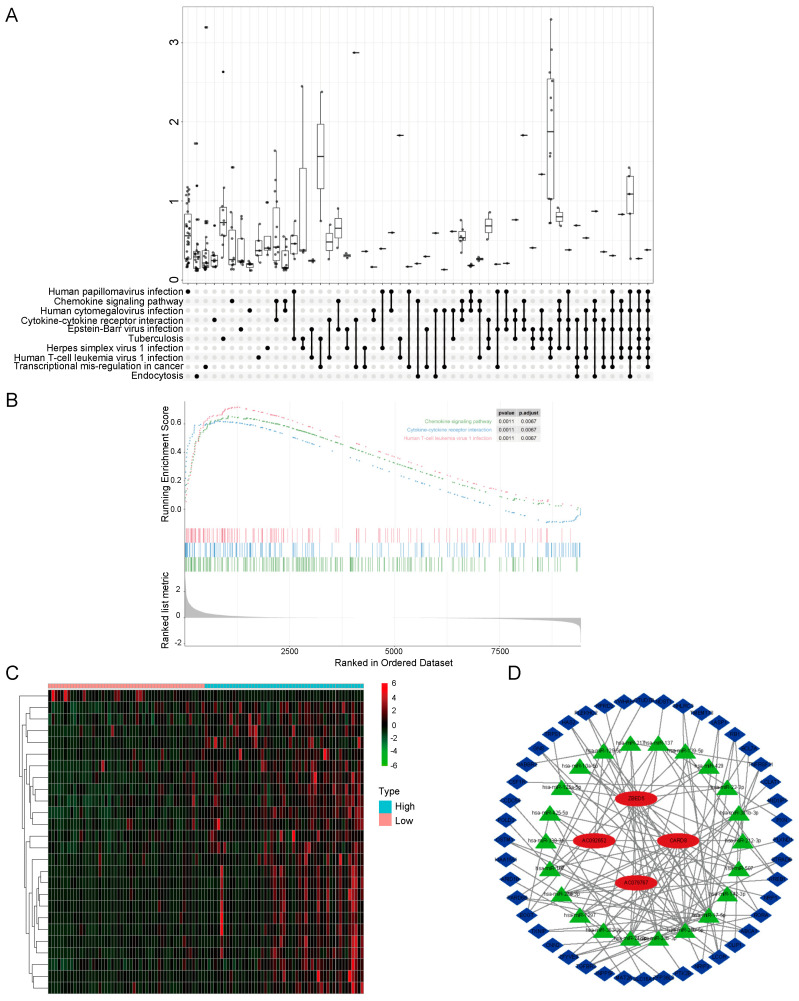
GSEA and ceRNA network construction. (**A**,**B**) The ssGSEA package was used for GO and KEGG enrichment analyses of all DEGs. The KEGG enrichment results are presented in the form of an upset plot (**A**). The top three terms in the GSEA were chemokine signaling, cytokine−cytokine receptor interaction, and Human T-cell leukemia virus 1 infection. (**C**) Differentially expressed lncRNA heat map for the validation set. (**D**) Based on the differentially expressed lncRNAs and mRNAs, a ceRNA network was constructed: green represents mRNAs, blue represents miRNAs, and red represents lncRNAs.

**Figure 3 biomolecules-13-00327-f003:**
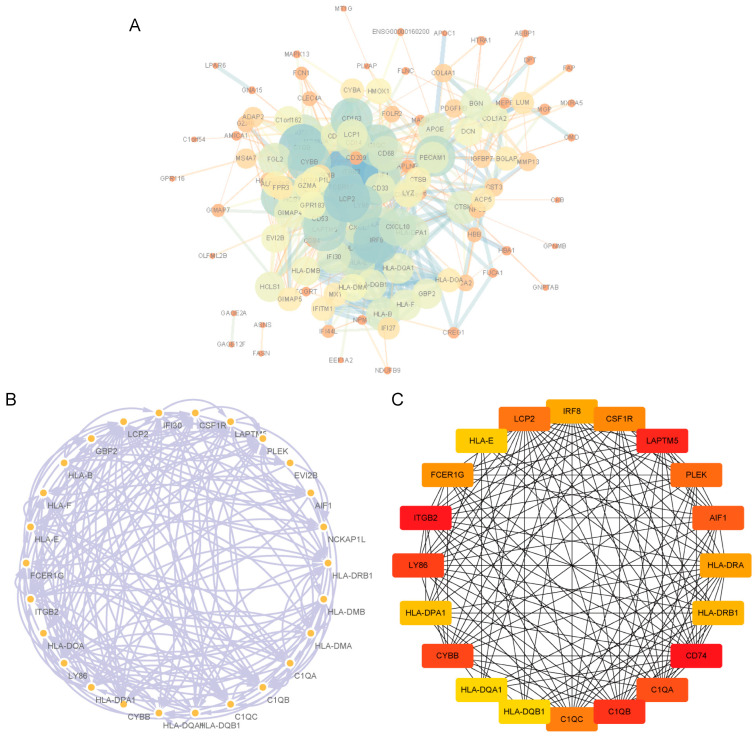
PPI network: (**A**) A PPI interaction network was constructed by using the molecular interaction list obtained from STRING database. (**B**) Hub gene sets obtained using MCODE of Cytoscape. (**C**) The top 20 hub genes obtained using cytoHubba of Cytoscape.

**Figure 4 biomolecules-13-00327-f004:**
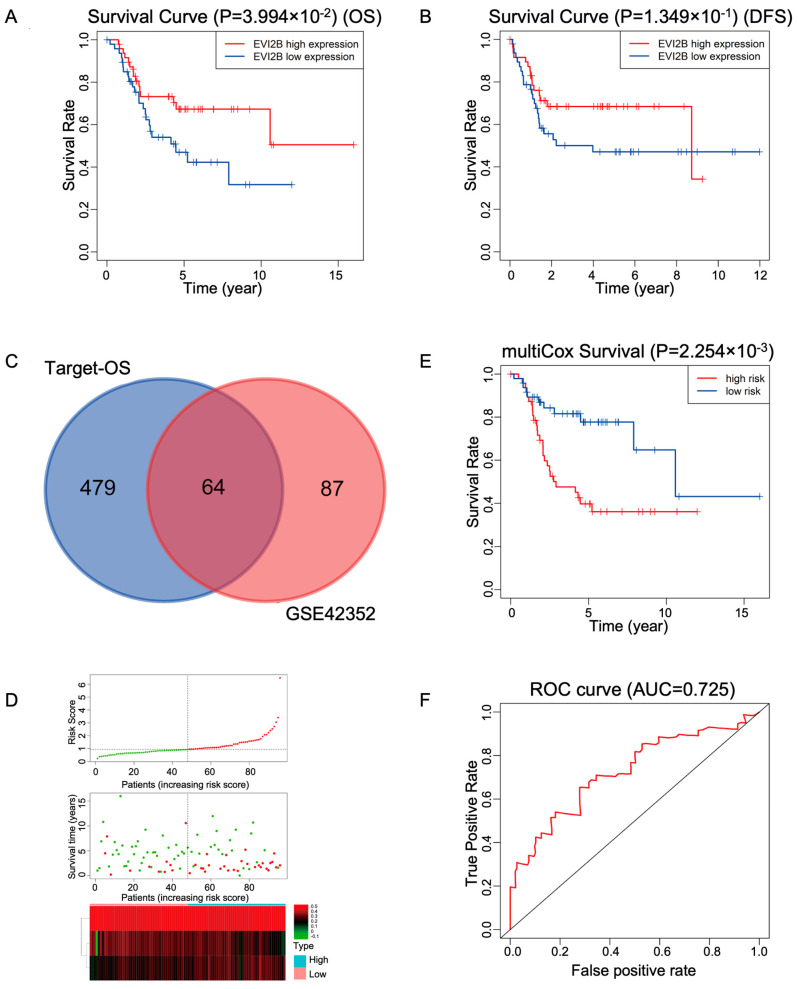
The prognostic value of EVI2B and the construction of a prediction model in osteosarcoma. (**A**) Effect of EVI2B on overall survival (OS) in osteosarcoma. (**B**) Effect of EVI2B on disease-free survival (DFS) in osteosarcoma. (**C**) Venn diagram showing the intersection between the differentially expressed genes in the test set with those in the validation set. (**D**) Risk curve of the three-gene set (**top**), survival state (**middle**), and risk heat map (**bottom**). (**E**) The three-gene set (*EVI2B, DOCK2,* and *CD33*) was a significant determinant prognosis based on overall survival in osteosarcoma, as determined by a multivariate Cox analysis. EVI2B and DOCK2 are highly expressed and CD33 is lowly expressed in low-risk group. EVI2B and DOCK2 are lowly expressed and CD33 is highly expressed in high-risk group. (**F**) ROC curve analysis of the accuracy of the model for the prediction of survival and prognosis in osteosarcoma.

**Figure 5 biomolecules-13-00327-f005:**
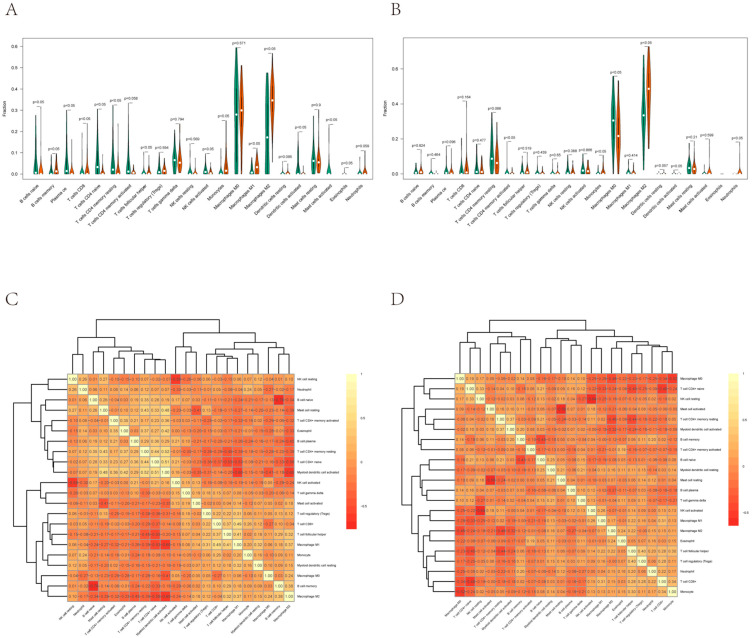
Differences in immune cell infiltration and correlations. (**A**,**B**) The CIBERSORT package was used to calculate differences in the infiltration of 22 kinds of immune cells between the high and low *EVI2B* expression groups in 103 osteosarcoma samples from the test set (**A**) and validation set (**B**). Green represents the low expression group and orange represents the high expression group in the violin plot. (**C**,**D**) Heat map of immune cell correlations in the test set (**C**) and validation set (**D**). Darker colors indicate a more significant correlation.

**Figure 6 biomolecules-13-00327-f006:**
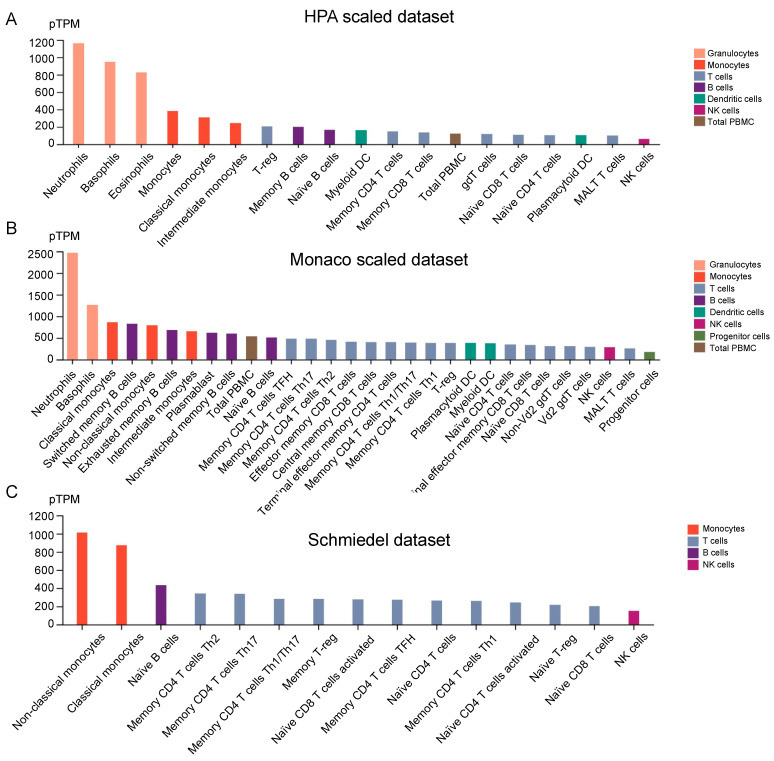
Protein expression level of EVI2B in different immune cells from HPA. (**A**) Protein expression level of EVI2B in the HPA scaled dataset. (**B**) Protein expression level of EVI2B in the Monaco scaled dataset. (**C**) Protein expression level of EVI2B in the Schmiedel dataset.

**Figure 7 biomolecules-13-00327-f007:**
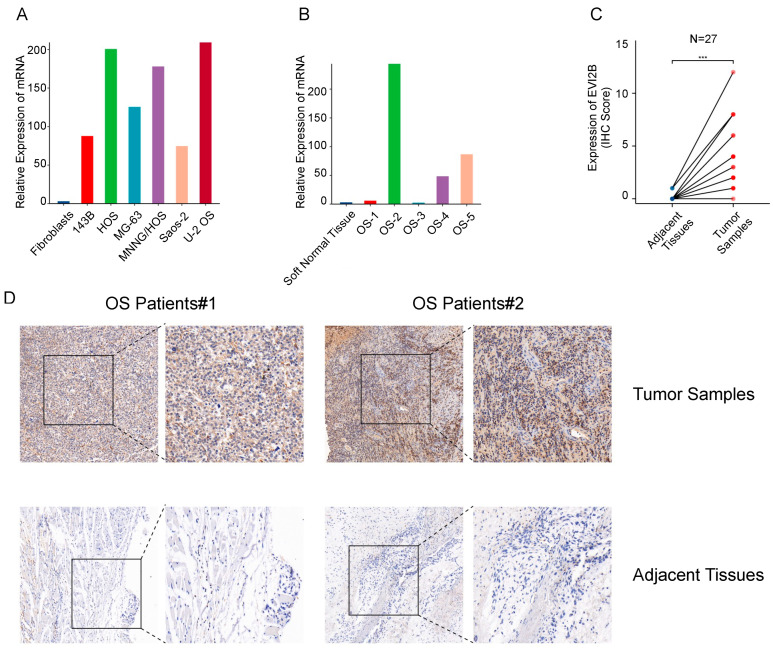
EVI2B expression in osteosarcoma cell lines and tissue samples. (**A**) Comparison of mRNA expression levels between different OS cell lines and fibroblast cells. (**B**) Comparison of mRNA expression levels in five OS tissue samples and adjacent normal tissues. (**C**) Immunohistochemical scores for 27 paired OS tumor tissues and adjacent normal tissues. (**D**) Representative immunohistochemical results for tumor and normal tissues in two patients with OS. *** *p* < 0.001.

**Table 1 biomolecules-13-00327-t001:** Multivariate cox regression analysis of three key genes.

id	coef	exp (coef)	se (coef)	z	Pr (>|z|)
DOCK2	−1.3304	0.2643	0.4440	−2.9962	0.0027
EVI2B	−8.7178	0.0002	4.6785	−1.8634	0.0624
CD33	1.0452	2.8439	0.6422	1.6275	0.1036

## Data Availability

The data supporting the findings of this study are available from the corresponding author upon reasonable request.

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
