# Peer review of "EVI2B Is a Prognostic Biomarker and Is Correlated with Monocyte and Macrophage Infiltration in Osteosarcoma Based on an Integrative Analysis"

_biomolecules, 2023, doi:10.3390/biom13020327_

Round 1

Reviewer 1 Report

Osteosarcoma (OS) is the most common malignant tumour of the bone. Current SOC involving surgery and extensive chemotherapy results in long-term survival in ~60% of patients. However, patients with local or metastatic recurrence have a 5-year survival rate of <30%. This has not improved in more than 3 decades. New research directions are needed to identify new treatment strategies. To do so, this study employed bioinformatic genome-guide analysis of datasets, including GSE4352 (test set) and The Cancer Genome Atlas Target-OS (validation set). This led to the identification of EVI2B as a gene whose high expression associates with good prognosis in OS. Expression of genes, and immune cell infiltrates were also evaluated in relation to high and low expression levels of EVI2B . An inverse correlation betweeb EVI2B expression and macrophage infiltration was observed. EVI2B mRNA and protein expression was analysed in OS and adjacent normal tissue, as well as in primary fibroblasts and OS cell lines, whereby a trend for higher expression in OS was observed.

The study is important as it validated the use of this bioinformatic approach as an avenue to identify potential mediators, prognostic, and predictive biomarkers in osteosarcoma.

The bioinformatic analysis was well designed and the observations derived from the analysis are reasonable in their majority.  The methods were also adequate for the samples in hand.

Following are important points to be addressed, in order to improve the quality of this manuscript:

1.     Whether the validation set was useful and how.

2.     Whether any of the non-coding RNAs and mRNAs identified here were previously identified by other groups using different approaches.

3.     It is unclear from the figure legend whether a high or low expression of the three-gene signature associated with poor prognosis. Please specify; the line colours in the figure are not sufficient.

4.     The hypothesis about the osteoclast-immune response relationship in the discussion. The explanation is unclear and needs experimental proof, otherwise it is too speculative. If the proof is in the study, the discussion will benefit from explaining it better.

5.     Why EVI2B expression was compared in OS cell lines vs. fibroblasts and not vs. osteoblasts?  primary human osteoblasts are commercially available.

6.     Why EVI2B protein was compared between OS and adjacent soft tissue and not between OS and normal bone?  Authors should also acknowledge the limitation in assessing differences in expression of EVI2B in OS vs. normal tissue when only 2 samples of normal tissue were assessed.

7.  The authors should acknowledge previous publications on EVI2B in osteosarcoma and discuss their findings in relation to these studies (doi: 10.21037/tp-22-402 and doi: 10.3892/ol.2021.12441).

Author Response

Dear reviewer,

We sincerely thank for your thorough and thoughtful comments that we have used to improve the quality of our manuscript. The comments are laid out below in blod fonts and specific concerns have been numbered. Our response is given in normal red font and changes/additions to the manuscript are given in red text. Also, As suggested, we have revised the manuscript carefully and performed additional descriptions that are described in new figures and in the revised text (in red).

We believe that the revised manuscript has addressed all the questions, and we hope that our manuscript is now suitable for publication.

Our point-by-point responses are included below.

With best regards

Tong Ji, M.D., Ph.D., Professor

Department of Oral and Maxillofacial-Head and Neck Oncology, Ninth People’s Hospital, Shanghai Jiao Tong University, School of Medicine, Shanghai

Ninth People’s Hospital, Shanghai Jiao Tong University, School of Medicine, 639 Zhizaoju Road Huangpu District, Shanghai, 200011, China

Point 1:  Whether the validation set was useful and how.

Response 1: We feel great thanks for your careful review work on our manuscript. We believe that the validation set used in this study is reliable and sufficient to support our results and conclusions. The validation cohort was derived from the Cancer Genome Atlas Target-OS. The two data sets used in this study are independent of each other, both of which are publicly available, and the number of samples contained in the two sets are similar (103 patients with OS in GSE42352 and 101 samples in Target-OS). We applied the same method used for the test set, 101 samples in Target-OS validation set were divided into high expression (n = 51) and low expression (n = 50) groups according to the median EVI2B value, and differentially expressed genes (DEGs) were identified between samples with high and low EVI2B expression in both the test and validation cohorts. So we believe the validation set was useful and the results validated by it were credible.

Point 2:  Whether any of the non-coding RNAs and mRNAs identified here were previously identified by other groups using different approaches.

Response 2: We feel great thanks for your thoughtful review work on our manuscript. Following your suggestion, we reviewed the bioinformation analysis on EVI2B in the Oncology field over the past 20 years. Notably, three studies have reported a relationship between EVI2B and the development of osteosarcoma. These studies are summarized as follows. In addition, we also search whether the non-coding RNA found in our analysis had been reported before in other osteosarcoma-related studies. Two related research were found. These studies are summarized as follows.

CARD8-AS1 was also identified as a low-risk cuproptosis-related lncRNAs in the study published recently(Yang et al., 2022). Transcriptome and clinical data of 86 patients with OS were downloaded from TCGA and GEO database. Pathway enrichment analysis, co-expression analysis, univariate COX regression analysis and least absolute shrinkage and selection operator (LASSO) regression analysis were performed to construct the risk prognostic model. Real-time quantitative PCR, Western blotting, and immunohistochemistry analyses verified the expression of non-coding RNAs in OS. AL645608.6, CARD8-AS1, AC005041.3, AC098487.1, and UNC5B-AS1 was upregulated in OS.

GAPLINC was identified as a predictor of poor prognosis and regulates cell migration and invasion in osteosarcoma(Liao et al., 2018). But that conclusion was not made by obtaining data from public databases and analysis. Based on the existing research results, the authors believed that GAPLINC may play a certain role in the occurrence and development of tumors (gastric cancer, colorectal cancer, and bladder cancer). Therefore, RT-qPCR was used to detect the expression of this lncRNA and a series of intervention experiments was conducted to identified that GAPLINC can affect the biological behavior of osteosarcoma cells.

Three studies confirmed the association between EVI2B and the prognosis of patients with OS  by means of bioinformatics analysis.

In the study conducted by Bingsheng Yang et al. in 2021, the Estimation of STromal and Immune cells in MAlignant Tumor tissues using Expression data (ESTIMATE) algorithm was applied to calculate the immune and stromal scores of patients with osteosarcoma based on data from The Cancer Genome Atlas database (TCGA). A metagene approach and deconvolution method were used to reveal distinct TME landscapes in patients with osteosarcoma. Bioinformatics analysis was used to identify differentially expressed genes (DEGs) associated with metastasis and immune infiltration in osteosarcoma, and a risk model was constructed using the DEGs with potential prognostic significance. Subsequently, gene set enrichment and Spearman's correlation analyses were used to delineate the biological processes associated with these prognostic biomarkers. GATA3, LPAR5, EVI2B, RIAM and CFH exhibited prognostic potential and were highly expressed in non‐metastatic osteosarcoma tissues validated by the IHC analysis results(Yang et al., 2021).

In another study, the authors obtained transcriptomic data related to osteosarcoma and osteosarcoma with metastasis from the Therapeutically Applicable Research to Generate Effective Treatment (TARGET) and The Gene Expression Omnibus (GEO) databases and identified the DEGs. A protein-protein interaction network analysis on potential key genes for osteosarcoma metastasis was also conducted. Then they conducted a Gene Ontology (GO) functional annotation analysis and Kyoto Encyclopedia of Genes and Genomes (KEGG) enrichment analysis to identify the core genes for prognosis, immune cell infiltration, and drug sensitivity. The risk prediction and prognosis models of metastasis were also constructed. Finally, they a metastasis prediction model with 5 genes (i.e., EVI2B, CEBPA, LCP2, SELL, and NPC2A)(Liang et al., 2022).

Tianyu Xie et al. obtained data of gene expression in HTSeq-FPKM type and the corresponding clinical information of 88 osteosarcoma cases from the TARGET database. The datasets (GSE21257) of 53 OS patients from GEO was used as a validation set. RNA-sequencing OS samples were categorized into high- and low- immune score groups with ESTIAMATE. Based on the immune score groups, 474 DEGs were acquired using the LIMMA package of R language. Subsequently, 86 DEGs were taken through univariate COX regression analysis, of which 14 were screened out by least absolute shrinkage and selection operator regression analysis. Multivariate COX regression analysis was performed to obtain 4 DEGs. Finally, EVI2B or CD361 gene was screened out via Kaplan-Meier analysis. In addition, CIBERSORT algorithm was used to evaluate the proportion of 22 kinds of tumor-infiltrating immune cells (TIICs) in OS. Correlation analysis revealed that the high expression level of EVI2B can elevate the infiltrated proportion of CD8+ T cells(Xie et al., 2022).

In addition to osteosarcoma, related bioinformation analysis have also been reported in other tumors. The studies included chronic lymphocytic leukemia (1), colorectal cancer (2), breast cancer (1), lung cancer (1) and melanoma (1), and five of them used data from public datasets. The above research is summarized below:

To identify new molecular markers for prognostication of chronic lymphocytic leukemia (CLL) patients, the authors analyzed cDNA arrays by using hierarchical clustering and standard statistic t-test on 34 CLL patients. They found expression differences in 78 genes compared to the reference tonsillar B lymphocytes. EVI2B was included in the cluster of genes(Aalto et al., 2001).

To detect circulating tumor cells (CTCs) for predicting early relapse of colorectal cancer (CRC) patients and analyze 15 candidate genes associated with CRC carcinogenesis. The authors constructed a weighted enzymatic chip array (WEnCA) platform including five prognosis related genes and analyzed the detection rate of WEnCA for circulating tumor cells (CTCs) in 30 clinically confirmed CRC relapse patients. Postoperative relapse was significantly correlated with gene overexpression, including EVI2B (p=0.001, OR=4.622)(Huang et al., 2012). In addition, another study, aimed to identify key modules and hub genes associated with the progression of CRC, obtained data of the patients with CRC from the Gene Expression Omnibus (GEO) database and assessed theses data by weighted gene co-expression network analysis (WGCNA). Gene Ontology (GO) and Kyoto Encyclopedia of Genes and Genomes (KEGG) enrichment analyses were also performed and several hub genes that regulate the mechanism of tumorigenesis in CRC were identified. The authors screened hub genes related to the progression of CRC authenticated by The Cancer Genome Atlas (TCGA) and Oncomine databases. Three hub genes (HCLS1, EVI2B, and CD48) were identified, and survival analysis was further performed(Yuan et al., 2020).

In another study, transcriptome data and matched clinical information of estrogen receptor (ER)-positive breast cancer (BRCA) were downloaded from the Cancer Genome Atlas. Immune (ISs) and stromal scores (SSs) of BRCA patients were calculated using the ESTIMATE algorithm. Inferred fractions of 22 types of infiltrating immune cells of BRCA were collected from the Cancer Immunome Atlas. Four hub genes (i.e., PLEK, CD53, EVI2B, and CD4) in this module showed significant association between their expression and ER-positive BRCA survival(Chen et al., 2021).

Tumor purity in 486 lung carcinoma tissues from TCGA-LUAD FPKM by using the “estimate” R package was measured. Lung carcinoma tumor mutation burden was calculated by analyzing TCGA single nucleotide polymorphism data. The immune cell proportion was also evacuated via the CIBERSORT method. Based on the tumor purity and lung carcinoma gene matrix, Weighted gene co-expression network analysis (WGCNA) was performed, and the tumor purity-related module was identified. Then, the functions of the factors were analyzed and the co-expressed factors related to clinical outcome and immunophenotype were screened. Finally, expression levels of these factors were measured at tissue and single cell levels(Bai et al., 2021).

Satoru Yonekura et al. integrated clinical data, mRNA expression data, and the distribution and fraction of tumor infiltrating lymphocytes (TILs) using The Cancer Genome Atlas (TCGA) and Gene Expression Omnibus (GEO) datasets (GSE65904 and GSE19234), and they found that EVI2B is a novel prognostic biomarker with IFN-γ associated immune infiltration in metastatic melanoma(Yonekura and Ueda, 2021).

Point 3:  It is unclear from the figure legend whether a high or low expression of the three-gene signature associated with poor prognosis. Please specify; the line colors in the figure are not sufficient.

Response 3: We sincerely thank the reviewer for careful reading. In Figure 4E, the patients were divided into groups “high risk” and “low risk” based on risk scores. Risk scores were calculated in Figure 4D after EVIB2, DOCK2, and CD33 were finally selected as prognostic biomarkers by multivariate Cox regression analysis (Table 1). As shown in the Table 1, the coefficients (coef) of EVIB2, DOCK2, and CD33 were -1.3304, -8.718 and 1.0452, respectively. To put it bluntly, patients with high expression of EVI2B or DOCK2 had a good prognosis, while those with high expression of CD33 had a poor prognosis. So, in the “high risk” group, EVI2B and DOCK2 were lowly expressed and CD33 was highly expressed. While in the “low risk” group, EVI2B and DOCK2 were high expressed and CD33 was low expressed. As you suggested, we have specified the detailed description in the figure legend.

In addition, as for another problem you pointed out “the line colors in the figure are not sufficient”, we found some distortion in the color of the figure after downloading manuscript for peer review generated by the submission website. We reviewed the original image uploaded and think that the colors are sufficient. So we didn't adjust the color. I hope you could forgive us.

Point 4:  The hypothesis about the osteoclast-immune response relationship in the discussion. The explanation is unclear and needs experimental proof, otherwise it is too speculative. If the proof is in the study, the discussion will benefit from explaining it better.

Response 4: We feel great thanks for your thoughtful suggestion on the point “osteoclast-immune response relationship” in the discussion. We have rethought this part of discussion, and as you said, the hypothesis about the osteoclast-immune response relationship is somewhat too speculative due to the lack of sufficient experimental proof and not enough reported research results to support. We consulted some literature and made some modifications to this part. The imbalance of bone homeostasis is closely related to the occurrence and development of osteosarcoma. Currently, some studies in the field of osteoarthritis have demonstrated that some bone marrow mesenchymal stem cell derived cells, including neutrophils, can affect the differentiation or function of osteoblasts or osteoclasts(Kitamura et al., 1995; Roodman, 2006; Sterrett, 1986). EVI2B was found highly expressed in neutrophils in osteosarcoma in our study. In view of the findings of this study and previous research, we speculated the dysregulation of EVI2B on immune cell may alter the bone homeostasis by affecting the generation or function of osteoclasts, thus affecting the corresponding response to therapy and prognosis. Given the results of the existing research, we believe this conjecture is reasonable and at the same time we must also admit that this conjecture needs to be further proved by experiments.

Point 5:   Why EVI2B expression was compared in OS cell lines vs. fibroblasts and not vs. osteoblasts?  primary human osteoblasts are commercially available.

Response 5: Thank you for your valuable advice. Osteosarcoma is defined histologically as a tumor of osteoid-producing cells, which often exist within an admixture of adipogenic, muscle, spindle, fibroblastic and chondroblastic cells. This microscopic phenotype has long fuelled the assumption that osteosarcoma arises from a multipotent mesenchymal precursor(Beird et al., 2022). Both fibroblasts and osteoblasts are derived from multipotent mesenchymal stem cells. Such homology in cellular origin provides some support for our selection. More importantly, when comparing human tumor samples with corresponding para-tumor tissues, we found that most normal cells in normal tissues were fibroblasts. Osteoblasts are mostly embedded in bone matrix, or exist in subperiosteum, with a small number. And that led to the fact that we only had fibroblasts as normal control cells in the samples from patients. In order to ensure the consistency of normal cells in cell lines and patient samples, we finally selected fibroblasts as normal control cells in vitro.

Point 6:  Why EVI2B protein was compared between OS and adjacent soft tissue and not between OS and normal bone?  Authors should also acknowledge the limitation in assessing differences in expression of EVI2B in OS vs. normal tissue when only 2 samples of normal tissue were assessed.

Response 6: We feel great thanks for your careful reading and thoughtful advice. As I mention in the Question 5, when comparing normal and tumor cells, it is difficult to use the small number of osteoblasts embedded in bone lacunae or subperiosteum as normal controls. Fibroblasts, which are homologous to osteoblasts(Beird et al., 2022), are abundant and readily available, so we ended up using fibroblasts located in soft tissues adjacent to tumor. As for the number of normal sample, it is possible that our statistical graph has caused this misunderstanding. We show two representative immunohistochemical staining results in Figure 7D. These two sample pairs were selected from a total of 27 sample pairs whose statistical results are shown in Figure 7C    . The IHC score of EVI2B in Figure 7C included 27 adjacent normal tissues and 27 corresponding tumor tissues. The EVI2B IHC scores in normal tissues are low and our statistical scores are integers from 0 to 15. This makes it look like there are only two points in the statistical chart of the Adjacent Tissue group.

Point 7:  The authors should acknowledge previous publications on EVI2B in osteosarcoma and discuss their findings in relation to these studies (doi: 10.21037/tp-22-402 and doi: 10.3892/ol.2021.12441).

Response 7: In the study titled ”Bioinformatics analysis of the key genes in osteosarcoma metastasis and immune invasion” (doi: 10.21037/tp-22-402), the authors compared the transcriptome data of osteosarcomas with or without metastasis from the TARGET and GEO databases. EVI2B, CEBPA, LCP2, SELL, and NPC2A were identified as potential key genes for osteosarcoma metastasis and macrophages was found as the predominant immune infiltrating cells in osteosarcoma metastasis(Liang et al., 2022). This article focuses on osteosarcoma metastasis. They found that patients with high expression levels of EVI2B had a better prognosis and EVI2B was associated with metastasis in osteosarcoma. We believe that this prediction model supports our conclusion it might be complementary to our prediction model. The authors screened and processed the data according to their primary focus: metastasis of osteosarcoma. Metastasis, although the most important factor affecting the prognosis of patients with osteosarcoma, is not the only one. Our results included all osteosarcoma samples data from the two selected databases. Combined with the results of the two studies, the predictive effect of EVI2B on the prognosis of osteosarcoma patients is not limited to metastasis, but also demonstrates in more detail that EVI2B is related to metastasis, a key prognostic factor. Moreover, we think that the analysis results of immune infiltration pattern in osteosarcoma are also mutually supportive. They identified macrophages as the predominant immune infiltrating cells in osteosarcoma metastasis and then they concluded that this may provide a new direction for the treatment of osteosarcoma. M0 macrophages were found significantly elevated and LCP2 and CEBPA were positively correlated with the level of macrophage infiltration. We also found similar results: M2 macrophages were significantly upregulated in the low EVI2B expression group. It has been shown that M0 and M2 macrophages are significantly associated with the prognosis of patients with osteosarcoma(Liu et al., 2021).

In another study (titled: Identification of prognostic biomarkers associated with metastasis and immune infiltration in osteosarcoma, doi: 10.3892/ol.2021.12441), the authors applied Estimation of STromal and Immune cells in MAlignant Tumor tissues using Expression data (ESTIMATE) algorithm to calculate the immune and stromal scores of patients with osteosarcoma based on data from TCGA database. Differentially expressed genes (DEGs) associated with metastasis and immune infiltration in osteosarcoma, and a risk model was constructed using the DEGs with potential prognostic significance. GATA3, LPAR5, EVI2B, RIAM and CFH exhibited prognostic potential and were highly expressed in non‐metastatic osteosarcoma tissues which was identified by IHC analysis results. As for the conclusion that EVI2B is a key prognostic gene of osteosarcoma, the two studies are consistent. We also applied IHC to verify the results of bioinformation analysis in samples.

In general, the results of the three studies corroborate each other.

Reviewer 2 Report

The authors discuss the possibility of defining EVI2B as a prognostic biomarker of survival in osteosarcoma (OS) and associating it with infiltrated immune cells. The analyses are mainly based on data analysis tools. In addition, they performed in vitro mRNA and protein analyses on cell lines and some patient samples. However, these results are not entirely clear regarding making EV12B a biomarker. All OS cell lines express this gene, so some types of OS cell lines do not express this gene.   What will be its phenotypic characteristics (metastatic capacity, for example)?  The author must explore different types of OS cell lines and their aggressiveness to establish that the expression of EVI2B is associated with the survival rate to answer that this gene is conferring aggressiveness, for example. When the authors explored this gene in human samples but cannot conclude what the meaning is in this sample of its expression, they must also associate these results with clinical data, tumor status, TNM data, metastasis or not, if the patient receives any therapy or chemotherapy. For example, the OS-1 and OS-3 samples showed low EV12B expression. What will be the explanation? 

For the latter, the expression of immune genes in the sample must be analyzed, or the degree of immune infiltration must be identified to be consistent with the manuscript's title and its in-silico data.

Minor consideration: 

In the introduction, they should expand more on the functionality of EVI2B and its association with the operating system.

All figures have a letter, they have a small font; It is impossible to determine the description. The problem is not the size of the figures, the problem is the size of the font, they must change it.

Author Response

Dear reviewer,

We sincerely thank for your thorough and thoughtful comments that we have used to improve the quality of our manuscript. The comments are laid out below in blod fonts and specific concerns have been numbered. Our response is given in normal red font and changes/additions to the manuscript are given in red text. Also, As suggested, we have revised the manuscript carefully and performed additional descriptions that are described in new figures and in the revised text (in red).

We believe that the revised manuscript has addressed all the questions, and we hope that our manuscript is now suitable for publication.

Our point-by-point responses are included below.

With best regards

Tong Ji, M.D., Ph.D., Professor

Department of Oral and Maxillofacial-Head and Neck Oncology, Ninth People’s Hospital, Shanghai Jiao Tong University, School of Medicine, Shanghai

Ninth People’s Hospital, Shanghai Jiao Tong University, School of Medicine, 639 Zhizaoju Road Huangpu District, Shanghai, 200011, China

Point 1: In the introduction, they should expand more on the functionality of EVI2B and its association with the operating system.

Response 1: We feel great thanks for your careful reading and thoughtful advice.  EVI2B was identified as one of eighteen new cell-surface molecules expressed on B cells. Although its function is not yet fully understood, EVI2B has been found as immune-related therapeutic indicator in several tumors(Aalto et al., 2001; Bai et al., 2021; Behring et al., 2021; Chen et al., 2021; Ferguson et al., 2022; Li et al., 2021; Yonekura and Ueda, 2021; Yuan et al., 2020; Zhang et al., 2022; Zjablovskaja et al., 2017). The identification and discovery of this new prognostic marker can empower clinical decision, and has great significance for clinical staging and predicting the outcome of clinical therapies. Accordingly, we have modified the introduction and added related description.

Point 2: All figures have a letter, they have a small font; It is impossible to determine the description. The problem is not the size of the figures, the problem is the size of the font, they must change it.

Response 2: We thank a lot for this valuable suggestion and we can’t agree more. Accordingly, we have modified the fonts of all the figures. Thank you very much for your reminding.

Round 2

Reviewer 1 Report

The responses to prior concerns are satisfactory. However, it is important that the information used to address these concerns (including references) is incorporated (in a succincter manner and without excessive detail) into the appropriate sections of the manuscript and this is tracked. The authors did this only for one section in the discussion, corresponding to lines 339-344. This section must revised as well, because the same statement appears in duplicate.

Author Response

Dear reviewer,

We sincerely thank for your careful review work. As suggested, we have revised the manuscript carefully. Our response is given in normal red font. Revisions made to the manuscript are marked up using the “Track Changes” function.

We believe that the revised manuscript has addressed all the questions, and we hope that our manuscript is now suitable for publication.

Our point-by-point responses are included below.

With best regards

Tong Ji, M.D., Ph.D., Professor

Department of Oral and Maxillofacial-Head and Neck Oncology, Ninth People’s Hospital, Shanghai Jiao Tong University, School of Medicine, Shanghai

Ninth People’s Hospital, Shanghai Jiao Tong University, School of Medicine, 639 Zhizaoju Road Huangpu District, Shanghai, 200011, China

Comments and Suggestions for Authors: The responses to prior concerns are satisfactory. However, it is important that the information used to address these concerns (including references) is incorporated (in a succincter manner and without excessive detail) into the appropriate sections of the manuscript and this is tracked. The authors did this only for one section in the discussion, corresponding to lines 339-344. This section must revise as well, because the same statement appears in duplicate.

Response: We feel great thanks for your careful review work on our manuscript. We are deeply sorry for our carelessness in the manuscript. The repeated paragraph has been corrected. In addition, according to your comments, we have also revised relevant sections in the discussion to make it looks more succincter.

Reviewer 2 Report

The auhtors have done all the changes suggested for imporiving the manuscript.  There is paragragh is the disccusion, line 342-344 that is repeted twices, should be corrected. Now, letteres from the image are Ok, but should be check in manuscript edition as will look.  

Author Response

Dear reviewer,

We sincerely thank for your careful review work. As suggested, we have revised the manuscript carefully. Our response is given in normal red font. Revisions made to the manuscript are marked up using the “Track Changes” function.

We believe that the revised manuscript has addressed all the questions, and we hope that our manuscript is now suitable for publication.

Our point-by-point responses are included below.

With best regards

Tong Ji, M.D., Ph.D., Professor

Department of Oral and Maxillofacial-Head and Neck Oncology, Ninth People’s Hospital, Shanghai Jiao Tong University, School of Medicine, Shanghai

Ninth People’s Hospital, Shanghai Jiao Tong University, School of Medicine, 639 Zhizaoju Road Huangpu District, Shanghai, 200011, China

Point 1: The auhtors have done all the changes suggested for imporiving the manuscript.  There is paragragh is the disccusion, line 342-344 that is repeted twices, should be corrected. Now, letteres from the image are Ok, but should be check in manuscript edition as will look. 

Response: We feel great thanks for your careful reading. We are deeply sorry for our carelessness in the manuscript. The repeated paragraph has been corrected and letterers from the image has been checked in manuscript edition.